# Oil Sands Wetland Ecosystem Monitoring Program Indicators in Alberta, Canada: Transitioning from Pilot to Long-Term Monitoring

Craig Mahoney *, Joshua Montgomery, Stephanie Connor ⓘD and Danielle Cobbaert

Alberta Environment and Parks, 9th Floor, 9888 Jasper Avenue, Edmonton, AB T5J 5C6, Canada;
joshua.montgomery@gov.ab.ca (J.M.); stephanie.connor@gov.ab.ca (S.C.); danielle.cobbaert@gov.ab.ca (D.C.)
* Correspondence: craig.mahoney@gov.ab.ca

**Abstract:** Boreal wetlands within the oil sands region of Alberta, Canada, are subject to natural and anthropogenic pressures, resulting in the need for monitoring these sensitive ecosystems to ensure their protection. This study presents results from Canada's pilot Oil Sands Monitoring (OSM) Wetland Program. This study is part of a project that seeks to assess and determine which of a selection of wetland indicators is suitable for identifying changes to wetland ecosystem "states" within a regional wetland monitoring program resulting from the effects of oil sands development. Specifically, this study seeks to identify indicators that can detect changes in a wetland ecosystem "state" using data from a 3-year pilot of the OSM Wetland Program and identify potential high-level oil sands-related pathways through which changes in states may occur, where appropriate. The monitoring data acquired during the pilot program are synthesized to identify preliminary trends and programmatic knowledge gaps, and future recommendations for an improved long-term "core" monitoring program are discussed. This study does not seek to attribute changes in wetland states measured via indicators to specific oil sands pressures but focuses on identifying those indicators that are sensitive enough to identify change over time. The results suggest that water quality, benthic invertebrates, and vegetation indicators can identify changes in wetland states over time, whereas wildlife indicators are inconclusive. Further, it is recommended that hydrometeorology data are acquired in parallel to other indicator data for contextualizing climate conditions. The findings from this work provide insights for developing and transitioning the OSM Wetland Program to a long-term effort, in addition to providing information for other regional wetland monitoring programs.

**Keywords:** wetlands; monitoring; water quality; aquatic ecology; hydrometeorology



## 1. Introduction

Wetlands in Canada's boreal region are diverse and vary in vegetation forms, soil types, hydrology regimes, and water chemistry [1]. These wetlands typically develop where the water table is at or near the surface, allowing water to settle and promoting the development of soil conditions for hydrophytic vegetation [2]. Boreal wetland ecosystems are critical for water security [3] and provide substantial economic value through a range of vital ecosystem services and functions, including carbon sequestration, flood attenuation and water storage, water filtration, wildlife habitats, and human recreation [4]. Moreover, boreal wetlands are highly valued by local Indigenous communities for culturally important foods, medicines, spiritual well-being, ecological values, and other culturally important uses [5–7].

Despite their known cultural and socioeconomic value, wetlands around the world are in a state of decline in terms of wetland area and quality [8]. This is a concern within Canada's province of Alberta, which exhibits one the greatest rates of boreal forest disturbance globally at approximately 78% (as of 2008) [9]. These high disturbance rates are common in the oil sands region (OSR) in the northeast of Alberta, where land cover

change is primarily attributed to natural resource extraction (oil and gas) and natural disturbances (e.g., wildfires and drying of peatlands due to climate change) [10]. In recognition of increasing human development pressures on wetland ecosystems across Alberta, the Government of Alberta (GoA) developed the Alberta Wetland Policy in 2013, which sets the strategic direction for wetland management and governance of all wetlands. The policy aims to conserve the values and ecosystem services wetlands provide society, prioritizing the conservation of wetlands with greater ecological value [11].

Canada's oil sands in northeastern Alberta contain approximately 10% of the proven oil reserves in the world [12]. The industry has expanded substantially since the first mines were developed in the late 1970s, with crude bitumen production (mined and in situ) totaling approximately 2.7 million barrels per day in 2021 [13,14]. Environmental monitoring programs were established to assess the potential effects of oil sands development on regional air quality, surface water and groundwater quality and quantity, and biodiversity (e.g., Alberta Oil Sands Environmental Research Program, Regional Aquatics Monitoring Program, Cumulative Environmental Management Association, Alberta Biodiversity Monitoring Institute, Wood Buffalo Environmental Association, Regional Groundwater Monitoring Network) [15]. However, scientific findings [16,17], criticism of existing environmental monitoring programs [18,19], and public perception of oil sands development on environmental health led to the Governments of Canada and Alberta developing the Oil Sands Monitoring (OSM) (formerly the Joint OSM) Program in 2012. The primary goal of Canada's OSM Program was to establish a world-class monitoring plan for the oil sands to provide assurance of the environmentally responsible development of the resource [20]. Oil sands industry operators are mandated by the Government through their operating approval conditions to contribute to the implementation of a regional wetland monitoring program to assess the cumulative effects of oil sands development on regional wetland ecosystems, in addition to project-specific on-lease wetland monitoring programs.

### 1.1. OSM Wetlands Program History

Under the OSM Program, a wetland monitoring technical advisory committee was established in 2017 to develop a study design and wetland indicators (and associated protocols) to determine the effects of oil sands development on regional wetland ecosystems. The wetland monitoring program was directed by the OSM Oversight Committee to address the following key questions: (1) Are changes occurring in wetlands due to contaminants and alterations to hydrological processes and land disturbances, to what degree are changes attributable to oil sands activities, and what is the contribution in the context of cumulative effects? (2) Are changes in wetlands affecting Indigenous health and well-being, culture use, and rights? Initial efforts to address these questions synthesized the scope of wetland research in the OSR, including the natural and anthropogenic pressures that impact wetlands and wetland characteristics sensitive to disturbances to develop a conceptual model of the study system [21]. These questions were the focus during the initial development of a conceptual model that was used to identify oil sands development pressures and anticipate wetland ecosystem responses, which formed a collection of hypotheses to guide the wetland monitoring program. Further, this conceptual model formed the foundation from which to identify wetland indicators that are known or predicted to be sensitive to oil sands development pressures, where an indicator is a characteristic "state" of the environment that quantifies the magnitude of stress, habitat characteristics, degree of exposure, or degree of ecological response to a pressure [22]. It has been noted that multiple pressures relate to changes in a single ecosystem state (e.g., vegetation abundance [21]); therefore, the ability to adequately measure changes in an ecosystem state is an important prerequisite to determine the source(s) of these changes (e.g., oil sands-related pressure(s)); the latter is deemed out of scope in this study.

The developed indicators were measured across a site network established during 2017 in the Athabasca Oil Sands Area of the OSR under a "pilot phase" of the OSM Wetlands Program. The site network established under the pilot phase of the OSM Wetlands Program

was based on the input of multiple stakeholders with an existing field-based presence in the OSR. For the ease of setup and rapid establishment of a wetland site network, the initial sites were colocated with existing sites where possible. Aligning with existing sites was deemed appropriate as the OSM Wetlands Program developed expertise in deploying appropriate instrumentation and monitoring the developed indicators. As a result of site colocation, the pilot phase of the monitoring program focused on bog, fen, and shallow open water wetland classes only, as these were the focus of existing projects in the region. Fen sites were colocated with a University of Waterloo project that focused on wetland hydrology, bog sites with a University of Villanova project that focused on the effects of contaminants on peatlands, and shallow open water (SOW) sites with amphibian and aquatic invertebrate groups lead by Environment and Climate Change Canada (ECCC). These sites were selected to answer research questions specific to each project, where the assessment of regional wetland conditions nor the effects of oil sands pressures on wetland ecosystems were not primary objectives. As the pilot phase of the program evolved, additional sites were added to the OSM wetland site network to better align with the goals of the OSM Wetland Program. It is acknowledged that a "purpose-built" site network is required to meet the goals of the OSM Wetland Program, specifically those related to the effects of oil sands operations. This is significant as the OSM Wetland Program transitions from its pilot phase to long-term monitoring.

### 1.2. Objectives

This study presents the preliminary results of the indicators identified from the wetland conceptual model developed by Ficken, Connor [21]. The indicator data acquired during a novel 3-year pilot phase of the OSM Wetlands Program across a suite of monitoring sites of various wetland classes are presented. Data analysis formed the foundation for making recommendations for the adoption and/or continued monitoring of high-value indicators as the program transitions to long-term monitoring. This study will inform long-term wetland monitoring at a regional scale to assess the effects of oil sands development, which is unique among other wetland monitoring programs in the region that focus monitoring efforts within oil sands lease boundaries. The objectives of this study are to:

i.　　　Identify high-level pathway–state relationships from oil sands development pressures;
ii.　　 Present pilot program monitoring data and preliminary high-level wetland indicator observations;
iii.　　Assess wetland indicator abilities to measure changes in the ecosystem "state", relating select indicator data to oil sands development pressures where appropriate;
iv.　　Discuss the limitations of the pilot study, proposed improvements, and recommendations for long-term monitoring efforts.

## 2. Materials and Methods

### 2.1. Study Site

Pilot wetland monitoring is focused on the OSR of Alberta, Canada (Figure 1). This is the world's third largest oil reserve and encompasses approximately 142,000 km$^2$ of various ecosystems in the boreal forest natural region of Alberta [13]. The region is characterized by short summers and long cold winters, with mean annual precipitation ranging from 478 mm to 495 mm around mining operations. The landscape includes vast upland mixed wood and coniferous forests with extensive, complex peatland (fen and bog), swamps, and SOW wetland systems in low-lying areas.

The study design comprised 22 monitoring sites established at wetlands assigned to fen (8), bog (7), and SOW (8) wetland classes (Table 1) (defined by the Alberta Wetland Classification System (AWCS) [23]), situated in an approximate 5000 km$^2$ area surrounding oil sands mining operations near the city of Fort McMurray (Appendix A: Table A1). Wetland monitoring sites incorporate existing research and monitoring sites from various existing studies, are characterized by an anthropogenic disturbance gradient from minimally to severely disturbed, and capture a range of hydrometeorological conditions. Of

note, language related to wetland characteristics follows the AWCS nomenclature [23]. The AWCS divides wetlands into "classes" based on hydrologic, biogeochemical, and biotic processes, which may be subdivided in "forms" based on dominant vegetation structure (e.g., wooded, shrubby, and graminoid).

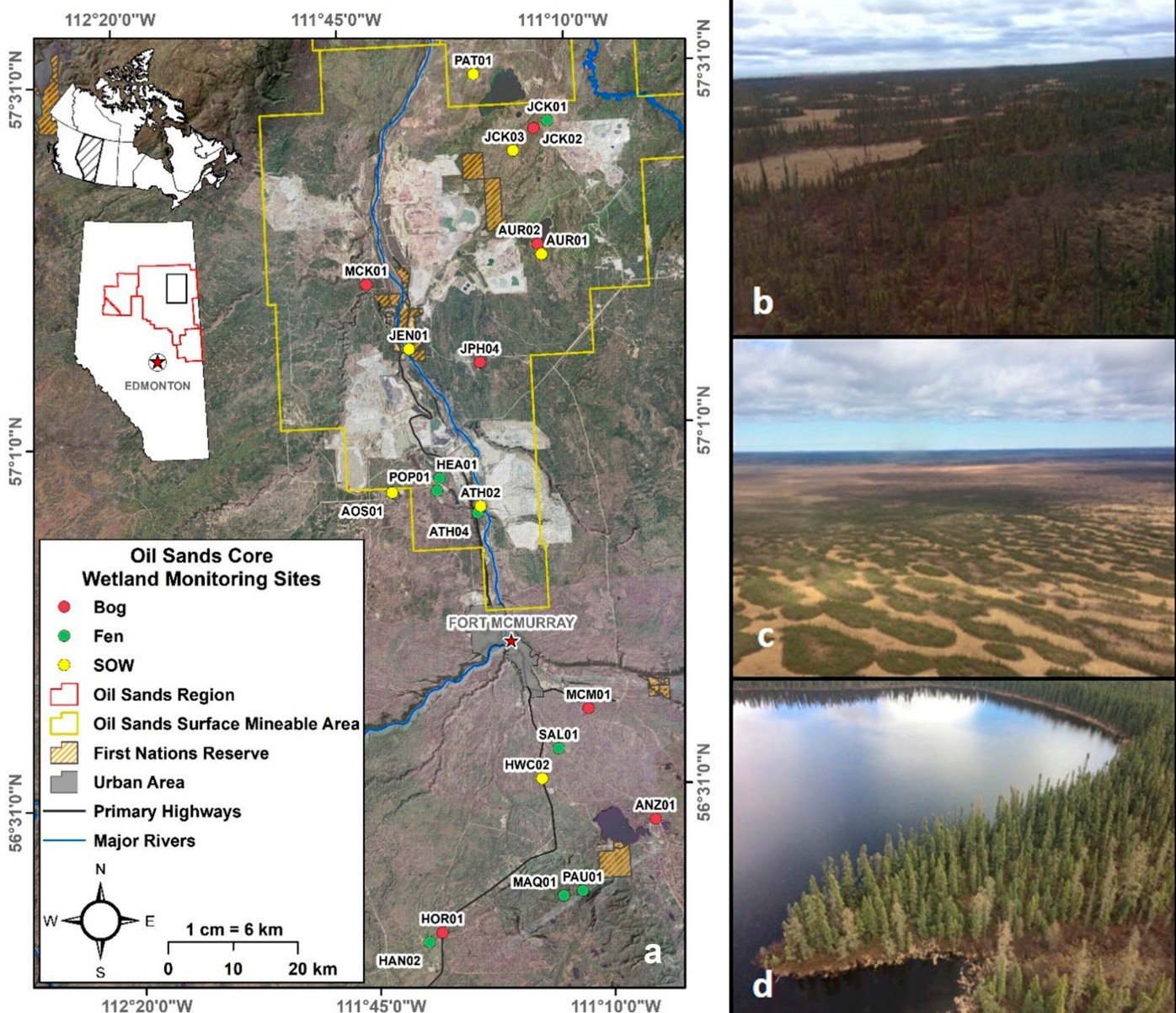

**Figure 1.** (**a**) Location of core wetland sites in the oil sands region of Alberta, Canada. Photos of each wetland type investigated in the study: (**b**) bog wetland (JCK02), (**c**) fen wetland (JCK01), and (**d**) shallow open water wetland (SOW) (JCK03). Map created using ESRI ArcMap. Photo credits: Joshua Montgomery.

## 2.2. Wetland Conceptual Model and Indicators

The developed wetland conceptual model is based on the Driver–Pressure–State–Impact–Response (DPSIR) Framework [24], which has been broadly adopted by member nations of the "Organization for Economic Co-operation and Development" [25] for environmental management and policy development. Under the DPSIR model, drivers exert pressures on a certain environment (e.g., addition of contaminants). Pressures cause the state of the environment to change (e.g., contaminant levels in biota). Impacts resulting from changes in ecosystem states can be social, economic, or environmental (loss of val-

ued species, such as caribou or muskrat). Impacts may lead to a societal response (e.g., improved management practices and government policies). Pressures are conceptualized as the mechanisms by which the drivers alter the state of the wetlands. State represents the condition and components of a wetland that may be defined by measurable abiotic and biological variables. Here, measurable states are known as "indicators" that can be observed and monitored under the program (e.g., the diversity of wetland flora or the quality of surface water).

**Table 1.** Landscape, soil, vegetation, and water characteristics of wetlands common within the oil sands region of Alberta.

| Wetland Class | Position in the Landscape | Soil and Vegetation | Water Regime and Chemistry |
|---|---|---|---|
| Fen | Flat to gentle slopes. Often part of wetland complexes. | >40 cm depth peat. Hummocky microtopography. High floristic species diversity. Wetland forms include wooded, shrubby, graminoid. | Minerotrophic (water inputs from surface runoff, groundwater, and precipitation). Surface and groundwater flow with near-surface water table. Generally freshwater but can be slightly brackish. Ranges from nutrient-poor to extremely rich. Wide range of pH (from neutral to slightly alkaline). |
| Bog | Flat elevated terraces. | >40 cm depth peat. Hummocky microtopography. Low floristic species diversity. Wetland forms include wooded, shrubby, open. | Ombrotrophic (water input primarily from precipitation). Low groundwater flow with a stable water table. Acidic pH and low nutrients. |
| SOW | Natural and anthropogenic topographic depressions or lake margins. | Mineral wetlands with <40 cm organic soil. Characterized by floating and aquatic vegetation in <2 m of open water. Wetland forms include floating or submersed aquatic, bare. | Minerotrophic (water inputs from surface runoff, groundwater, and precipitation). Permanent open water bodies in the oil sands region with dynamic seasonal water levels. Nutrient-rich freshwater or saline. Typically, neutral pH. |

Wetland indicators for the pilot program were selected based on three priority oil sands development pressures—land disturbances, hydrological disturbances, and contaminants. These indicators were developed in accordance with the principles of the OSM Program as follows: holistic and comprehensive, scientifically credible, risk-based, interpretable, cost-effective, and culturally and biologically relevant [26]. Indicator selection was performed in collaboration with multiple stakeholders, including representatives from Government, industry, nongovernment organizations, academia, and local Indigenous communities. No single indicator will possess all the desirable properties so a suite of complementary indicators spanning physical, chemical, and biological components was chosen. To protect against unknown impacts or ecological "surprises", a broad suite of indicators was identified that collectively constitute environmental conditions and "effects" (i.e., biological and ecological impairment, long-term changes) to composition, structure, and/or function of the disturbed ecosystem [27].

A DPSIR model was built around the selected indicators, illustrating conceptual pressure–pathway–state response under the three identified priority pressures. This represents an extension on the work of Ficken, Connor [21], indicating pathways and summarizing the predicted indicator (state) responses to these pressures.

*2.3. Data Analysis*

Priority wetland indicators were assessed to examine intra- and interannual variability between site variability (contingent on data availability) and broad data trends. Due to the short and inconsistent time series of data (up to 3 years) collected across indicators, assessment of whether individual indicators are valuable to the OSM Wetlands Program was based on known and/or predicted responses, informed by literature. Processing steps

are described for each indicator stratified under the following themes: hydrometeorological, water quality, benthic invertebrates, wildlife, and vegetation composition. All data analysis and figure production were performed using the "R" programming language. All data are publicly available from http://osmdatacatalog.alberta.ca/.

Two levels of analysis were performed based on the suitability of sites; indicators were assessed at either a high level (to identify change over time) or an increased level of detail where broad-scale relationships with oil sands pressures were explored. The latter was performed for SOW sites, where water quality and benthic invertebrate indicator measurements were concentrated, and the former was applied for the analyses of wildlife and vegetation data.

### 2.3.1. Hydrometeorological Data

Air and water temperatures, precipitation, soil moisture, and depth to water table data were synthesized and summarized to demonstrate the presence of natural climate-related variability on intra- and interannual scales. Data were recorded using Onset HOBOware sensors (see Appendix A: Table A2 for further reading on sensor specifications) installed on a 2 m tripod, recorded at 1 h nominal intervals, and summarized along a daily time series during postprocessing. Air and water temperatures, soil moisture content, and depth to water table were summarized as daily mean values, whereas precipitation was summed to yield daily totals. Daily summaries of each variable ensured identical timestamps existed for unique within-site data records that facilitated comparisons between site data records. To mitigate issues associated with differing data record lengths, all records were standardized from 1 May to 31 October, where data gaps were filled with null values. This standardized record length precedes and exceeds anticipated setup or takedown times for any year and, therefore, acts as endmember for current and future acquisitions.

The summary of raw data along a daily time series facilitated a means of quality assurance and quality control (QAQC) for records with irregular recording intervals or duplicate measurements. Where data were erroneously duplicated, the process of time-series standardization facilitated the removal of these spurious records. Additional QAQC criteria related to expected limits related to each variable were also developed and applied. For example, air temperatures are not expected to exceed ±40 °C during summer across the region; nonconforming records are, therefore, excluded. QAQC rules are applied during postprocessing and can, therefore, be modified depending on observed conditions throughout the year.

Daily (QAQC'd) data records were further summarized to monthly means as a function of wetland class for 2018 and 2019, facilitating the assessment of trends within and between wetland classes. Soil moisture and depth to water table data are restricted to 2019 only because 2018 records are largely incomplete due to instrument malfunction. Similarly, daily (QAQC'd) precipitation records (2018 and 2019) were summarized to yield monthly totals for each wetland site. Data were organized as a function of each wetland class for each year to highlight potential intra- and interannual variability between sites and wetland classes.

### 2.3.2. Water Quality

Samples for analysis of water chemistry were collected from shallow open water and open fen wetlands. Samples were collected in mid to late summer (constituting one sample per year per site) after the effects of snowmelt and spring flooding had subsided. Grab samples were collected from mid-depth at a single location within each wetland following standard collection protocols for monitoring of surface waters in Alberta [28]. Samples were collected from an area of the water column free of debris and excess vegetation and were collected prior to other sampling activities (e.g., benthic invertebrate sampling) to minimize disturbance of the vegetation and sediment. Water samples were analyzed for a modified suite of water quality parameters outlined in the plan for Phase I monitoring in the Oil Sands Monitoring Program [29]. Briefly, this included routine parameters (e.g., pH, conductivity), major ions, nutrients and carbon, and trace metals, including

mercury and polycyclic aromatic compounds (PACs). Water samples were preserved on site (as necessary) and analyzed by accredited commercial laboratories following standard detection methods with strict QAQC procedures. Additional detail on individual water quality sample analyses methods (including federally defined methods codes), commercial vendors, and detection limits are documented in spreadsheets available from the OSM data catalogue (http://osmdatacatalog.alberta.ca/dataset/wetland-surface-water-quality). All analyzed water parameters were assessed against established limits (where available) identified in Alberta's surface water quality guidelines [30] and the Canadian Council of Ministers of the Environment (CCME) Water Quality Guidelines for the Protection of Aquatic Life (PAL) (https://ccme.ca/en/resources/water-aquatic-life). All water quality parameters (excluding sulphur) were correlated (Pearson's correlation coefficient) with shortest distance to oil sands mines (based on industry-reported human footprint data for 2017 [31]); sulphur was assessed as a function of nearest upgrader stacks in 2017 [32].

### 2.3.3. Benthic Invertebrates

Benthic invertebrate samples were collected at shallow open water wetlands (and open fens) in mid to late summer over a three-year period (2017 to 2019). A single representative sample was collected from each site following the Canadian Aquatic Biomonitoring Network (CABIN) protocol for wetland habitats [33]. Sampling consisted of a 2 min traveling sweep with a 400 μm kick net through the submerged and emergent vegetation in the littoral zone of the wetlands. Excess vegetation in the sample was rinsed on site, and the debris was discarded. Samples were preserved immediately in 95% ethanol and stored under dark, cold conditions.

Benthic invertebrates were identified and enumerated by certified taxonomists at commercial taxonomic laboratories following the CABIN standard lab protocol [34], including all QAQC procedures, which are documented in spreadsheets available from the OSM data catalogue (http://osmdatacatalog.alberta.ca/dataset/wetland-benthic-invertebrates). Samples were subsampled, and a minimum of 5% of each sample was processed. If after 5%, the sample did not include 300 individuals, then subsampling continued until 300 individuals were identified. Data are presented here at lowest practical level (LPL), usually genus. Predator richness, an additional invertebrate metric known to vary in the OSR between wetlands with and without oil sands process materials [35], is also presented. The correlation between benthic invertebrate community data with human footprint density (within 500 m buffer; see [36]) and shortest distance to oil sands mines (based on industry-reported human footprint data for 2017 [31]) was assessed as a proxy for the influence of oil sands pressures, reporting Pearson's correlation coefficient.

### 2.3.4. Wildlife

Motion detection cameras (Reconyx Hyperfire 2, WI, USA) and autonomous recording units (ARUs; Song Meter SM4, MA, USA) were deployed at each site to monitor wildlife (i.e., mammals, birds, amphibians); see Appendix A: Table A2 for further reading on sensor specifications. Cameras were deployed year-round for the capture of primarily mammals and some birds, while ARUs were deployed for the open water season for the capture of songbirds and amphibians. Camera data for 2018 comprised captures made between June 2018 and May 2019 (excluding MAQ01 in 2018), whereas 2019 data were defined by captures made between May and October (excluding ATH02 and MAQ01). Cameras captured a single image with each movement trigger, as well as a programmed daily capture at midnight. All images captured during the deployment period were analyzed manually, and presence and abundance of wildlife was recorded. Individuals were identified to species where possible.

Autonomous recording units were programmed to capture daily peak vocalization periods for amphibians and songbirds following the standard SM4 schedule outlined by the Bioacoustic Unit at the Alberta Biodiversity Monitoring Institute (ABMI; [37]). Capture times were centered around midnight, sunrise, midday, and sunset. Capture intervals

varied, where a total of 56 min were recorded per 24 h period. Four day and night periods were chosen at random for analysis from within the deployment period with a limit set in early July, and all audio recordings from those four dates were analyzed. After mid-July, the probability of detecting a species drops significantly due to less vocalization. Therefore, recordings were chosen from a period of peak breeding season and optimal weather.

### 2.3.5. Vegetation

Sampling methods followed standard operating procedures (SOPs) that characterize wetland vegetation using a series of transects and plots at each wetland with emphasis on wetland margins and transition zones where vegetation may be the most sensitive to impacts from changes in hydrology or nutrients cycles [38]. Use in remote sensing applications for scaling information to regional scales or calibration and/or validation of wetland classes, types, and extents were also considerations of data acquired using vegetation SOPs. A series of three transects (except MCM01, where two was found sufficient), comprised five 1 m quadrats every 5 m along each transect, were established at different locations near the upland–wetland transition to best capture proximal influences around the wetland. Transects began in an upland region adjacent to the wetland extending toward the center of the wetlands to best capture sensitive transition zones. Vegetation species data, identified as a priority indicator by the conceptual model, were acquired by a vascular plant expert at each quadrat and synthesized to species richness.

## 3. Results

### 3.1. Wetland Conceptual Model and Indicators

A critical assessment of each indicator expands on the initial findings by Ficken, Connor [21], identifying pathways (Figure 2) and suggesting the observed/predicted response of each indicator in relation to each priority oil sands development pressure (Table 2).

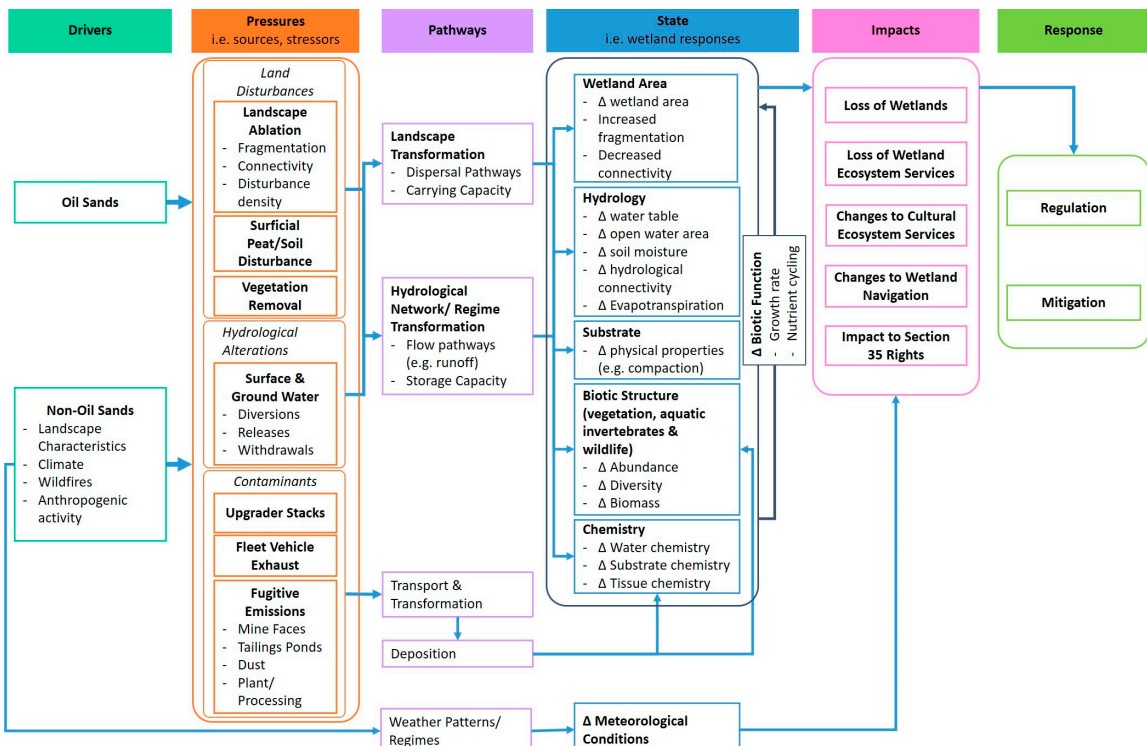

**Figure 2.** The DPSIR (Driver–Pressure–State–Impact–Response) Framework applied to the OSM Wetland Ecosystem Monitoring Program, including high-level pathways through which pressures exert influence on ecosystem states. Note, Δ means 'change in' an indicator.

**Table 2.** Oil sands (OS) wetland monitoring indicators: wetland indicators that are known or predicted to be sensitive to OS development address concerns and values of local Indigenous communities and are consistent with the principles of the Oil Sands Monitoring Program, i.e., risk-based, scientifically credible, holistic, comprehensive, repeatable, and comparable [26]. Predicted responses to oil sands development pressures are focused on land disturbance, hydrologic alteration, and contaminants.

| Wetland Indicators | Indicator Rationale | Predicted Responses to Priority Oil Sands Development Pressures |
|---|---|---|
| Wetland area. (Changes in wetland area, fragmentation, loss of connectivity). | Wetland area status and trends are critical indicators of wetland health and condition [8]. Northern Alberta has one of the fastest rates of land disturbance [9]. Local Indigenous communities are concerned about land use change. | Land disturbances result in direct wetland loss, increased fragmentation, and decreased connectivity [39,40]. Changes in the habitat and abundance of traditional plant areas, wetland-reliant species at risk, and biodiversity [41]. |
| Meteorology. (Precipitation, temperature, relative humidity, and wind speed and direction). | Contextualize the influence of local climate on wetland hydrological conditions versus anthropogenic development. Contextualize wetland hydrological functioning as related to 10–15-year wet–dry climate cycles that characterize the OSR [42]. | Climate change is predicted to affect the duration of wetland hydrological connectivity in the region [43]. |
| Hydrology. (Water table depth, soil moisture levels) | Wetlands provide hydrological ecosystem services [44]. Hydrology is sensitive to local land disturbances and anthropogenic hydrological alterations [43,45]. Water table position and open water area are proxies for assessing change in wetland function [46]. Local Indigenous communities are concerned about access routes to harvesting and occupancy sites. | OS water management may result in abnormal water table positions [45], resulting in terrestrialization and changes to runoff. Localized infrastructure development may obstruct wetland natural subsurface flow, changing hydrodynamics [47,48]. |
| Surface water quality. (Full suite of OS SWQ parameters of concern for shallow open water wetlands; reduced suite of parameters in peatlands). | SWQ parameters provide a measure of aquatic habitat condition relative to the needs of flora and fauna (e.g., habitat, drinking water, etc.) [29]. Deposited contaminants can be transported large distances through the hydrological network (e.g., surface water and/or groundwater). Multiple OS contaminants can modify wetland function [49–51]. | Directly deposited or transported contaminants may cause eutrophication/nitrification [49,50]. Contaminant concentrations may change in relation to established guidelines [30,52,53]. Potential to change specific conductance and pH [54]. |
| Sediment quality. (Shallow open water wetlands only; full suite of OS sediment parameters of concern). | Sediments are contaminant sinks and a major exposure route for plants, invertebrates, amphibians, and birds. | Contaminant concentrations may change in relation to established guidelines (e.g., CCME PAL Guidelines). Shallow lake sediments near the OS mining center are enriched in vanadium and nickel [55]. |
| Vegetation. (Community composition and structure; culturally important plants; high disturbance indicator species; obligate wetland species). | Plant communities are sensitive to natural and anthropogenic drivers. Culturally important plants are a proxy of wetland health and change [6]. Local Indigenous communities are concerned about changes in vegetation communities reducing biodiversity [5,41]. | High disturbance indicator species are more common in wetlands nearer to land disturbances [56]. Change in vegetation community composition resulting from contaminant deposition. |

**Table 2.** *Cont.*

| Wetland Indicators | Indicator Rationale | Predicted Responses to Priority Oil Sands Development Pressures |
|---|---|---|
| Benthic invertebrates. (Shallow open water wetlands only; community composition). | Benthic invertebrates are small, aquatic organisms commonly used to assess the environmental condition of freshwaters (rivers, lakes, wetlands) across Canada [57]. | Benthic invertebrate communities are sensitive to the extent of land disturbance in wetland buffers and associated changes in surface water quality. |
| Wildlife. (Remote cameras and acoustic recorders). | Wildlife are sensitive to land disturbances and human activity. Local Indigenous communities are concerned that fewer wildlife are using wetlands [41]. | Potential for negative ecological and socioeconomic impacts on wildlife due to anthropogenic activity [58]. Increased human noise and activity has potential to reduce wildlife habitat and presence [59,60]. |

### 3.2. Hydrometeorological Data

Select hydrometeorological variables are summarized over acquisitions made at all 2019 sites and indicate variability exists within and between the wetland classes (Figure 3). Data records indicate high-level trends as a function of wetland class over time (Figure 3). For example, the air temperature of peatlands is consistently cooler than SOW wetlands, where bogs and fens exhibit lower minimum, median, and maximum temperatures. These trends are amplified in water temperature observations, where mean SOW temperatures are approximately 7 °C and 11 °C warmer than fens and bogs, respectively. Similarity exists between air and water temperatures for bog, fen, and SOW classes; specifically, warming and cooling trends follow similar trajectories over the observation period between May and October. A seasonal "tipping point" is observed from July to August, where air temperatures in August are consistently lower than in July. This trend reversal is noted at different times for water temperature, depending on the wetland class. SOW classes follow a similar trend to that observed in air temperature, whereas fens exhibit a trend reversal a month later. Bogs indicate no trend reversal.

Soil moisture varies between wetland classes, where fens exhibit the highest mean moisture contents of all classes ($0.55 \pm 0.01$ m$^3$m$^{-3}$; mean $\pm$ standard deviation), followed by bogs ($0.41 \pm 0.08$ m$^3$m$^{-3}$) and SOW ($0.31 \pm 0.02$ m$^3$m$^{-3}$). Bogs exhibit the most variability of all wetland classes, whereas fens and SOW wetlands exhibit similar, low variability ($0.01$ m$^3$m$^{-3}$ difference in standard deviation). The depth to water table and associated variability are greatest in SOW wetlands ($0.68 \pm 0.07$ m) compared to bogs ($<0.01 \pm 0.06$ m) and fens ($-0.01 \pm 0.04$ m) (Figure 3d). Variability is noted within sites between years (monthly precipitation; Figure 4). In general, larger precipitation events were observed earlier in 2018 (July) compared to 2019, where larger events were observed later (August). Moreover, the mean ($\pm$standard deviation) precipitation for bogs, fens, and SOW wetlands varies between 2018 ($40.4 \pm 39.5$ mm, $39.0 \pm 52.1$ mm, and $42.6 \pm 46.3$ mm, respectively) and 2019 ($46.0 \pm 47.5$ mm, $49.8 \pm 50.8$ mm, and $44.3 \pm 43.2$ mm, respectively).

### 3.3. Water Quality

A subset of three years (2017 to 2019) of water quality parameters (Table 3) potentially related to oil sands development activities is detailed in Figure 5. Total nitrogen (Figure 5b) is variable among sites but typically consistent across years. Nitrogen does not correlate with distance to the upgrader or nearest oil sands surface mine but is consistently elevated at site AOS01. Sulfate (Figure 5a) is elevated at sites near to upgrader stacks (JEN01, AOS01) but did not exceed the sulfate guidelines (309 mg L$^{-1}$ in hard water; [30]) and is largely below the detection limit among the rest of the monitoring network. Base cations decreased in concentration with increasing distance to the oil sands mine (Figure 5c), excluding HWC02 in all available years (2018, 2019). There was no relationship between trace metal concentrations and distance to the oil sands surface mine, but metal concentrations were consistently elevated at site AOS01 (Figure 5d–h), including two exceedances of the Protec-

tion of Aquatic Life (PAL) guidelines for dissolved aluminum in 2017 and 2018 (Figure 5d). Additionally, site AOS01 exceeded chronic exposure guidelines for methylmercury in 2018 and 2019 but did not exceed total mercury guidelines in any year. There were no exceedances of the PAL guidelines for nickel or vanadium. The concentration of alkylated polycyclic aromatic hydrocarbons (Alk-PAHs) did not vary with the distance to the nearest oil sands mine (Figure 5i).

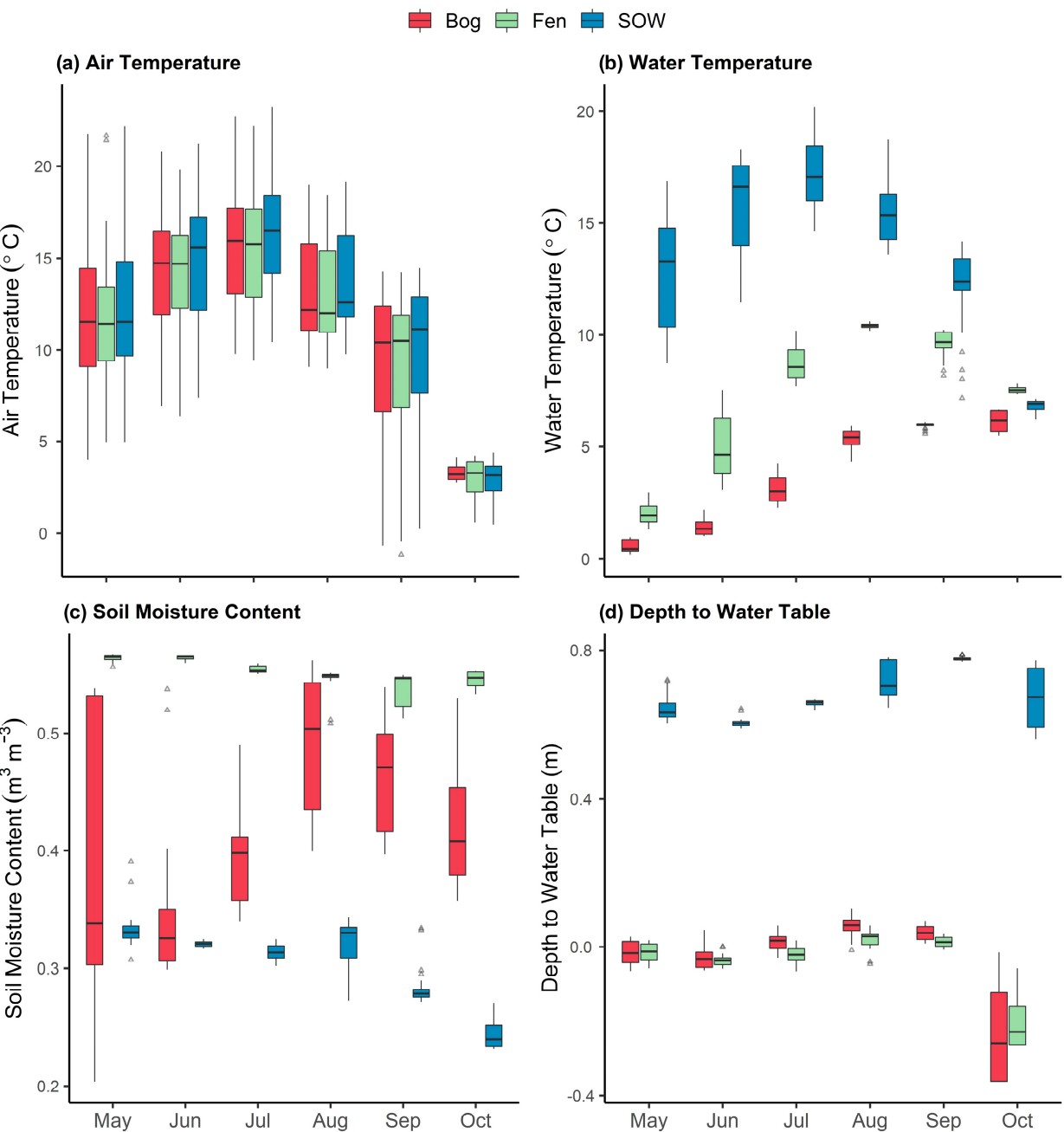

**Figure 3.** Summary boxplots illustrating the variability and broad trends of selected meteorological and hydrological variables recorded from all 2019 study sites: (**a**) air temperature, (**b**) water temperature, (**c**) soil moisture content, and (**d**) depth to water table (positive depths indicate the water table is above the surface). Variables are stratified as a function of Alberta Wetland Classification System "class": bog, fen, and shallow open water (SOW).

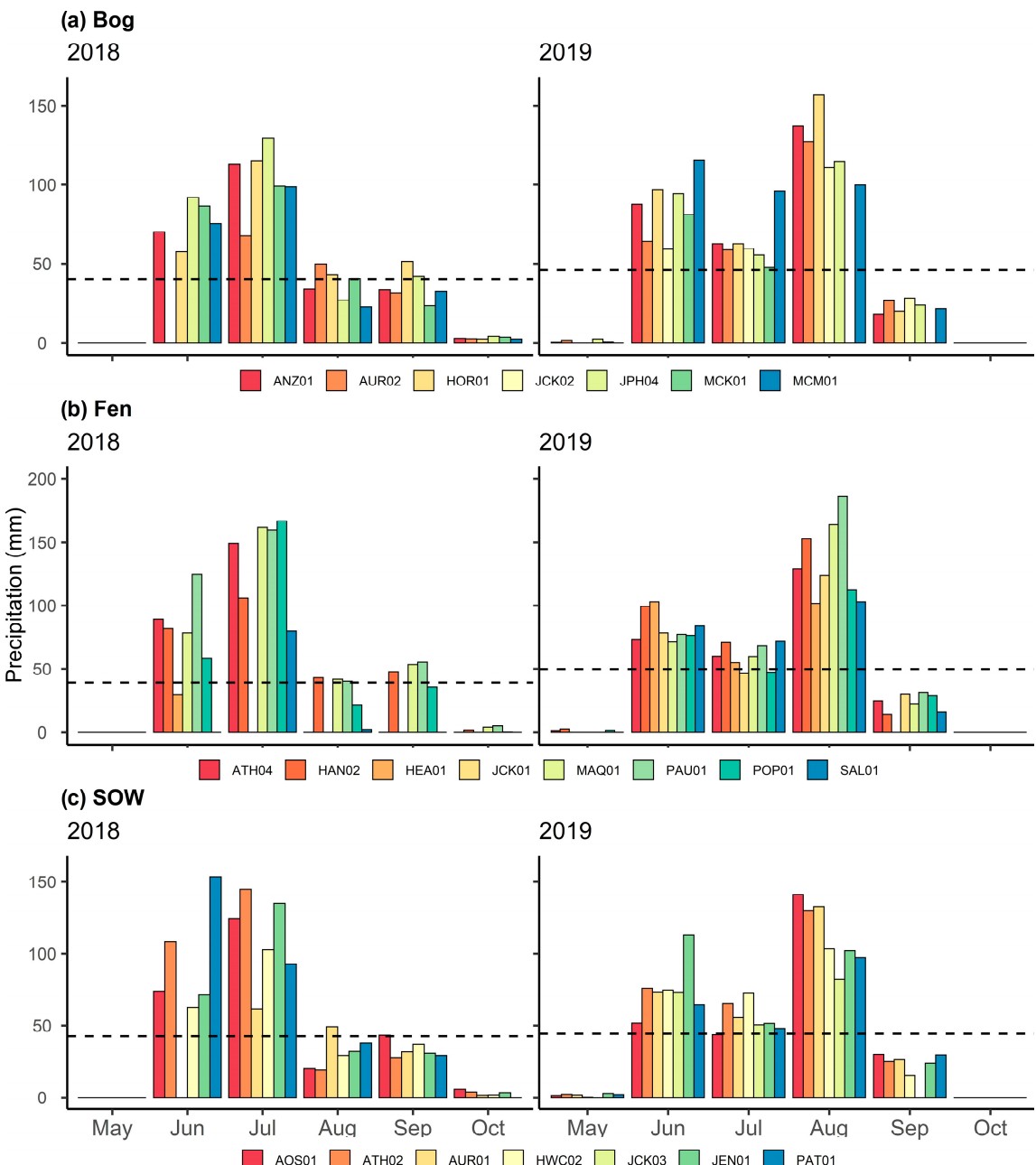

**Figure 4.** Monthly summation of precipitation records for 2018 and 2019 at individual wetland sites stratified as a function of wetland class: (**a**) bog, (**b**) fen, and (**c**) shallow open water (SOW). Gaps in data records are due to a lack of records from instrument failure or variable record start and end times. Dashed line represents mean precipitation for the data collection period within each year.

**Table 3.** Water quality parameters of interest potentially related to oil sands development.

| Parameter | Potential Oil Sands-Related Source | Importance |
|---|---|---|
| Total Nitrogen | Industrial emissions (stack, fleet) of $NO_X$; emissions of $NH_3$ from tailings [61]; microbial fixation [62]. | Eutrophication of low-nutrient habitats (i.e., bogs and poor fens); shift from bryophyte-dominated to vascular-dominated communities [50]. Potential ammonia toxicity to fish and other aquatic life but dependent on pH and temperature [53]. |
| Sulfate | Industrial emissions of $SO_2$ and $H_2S$ [51]. | Acidifying deposition [63]; sulfate toxicity is hardness dependent [30]. |

**Table 3.** *Cont.*

| Parameter | Potential Oil Sands-Related Source | Importance |
|---|---|---|
| Σ Base Cations | Deposition of fugitive dust from surface mining/surficial erosion [54]. | Some evidence of neutralizing acid deposition [64,65]; potential to increase pH in bogs. |
| Total and Methylmercury | Industrial emissions [66]; global deposition [67]; in situ fixation [68]. | Bioaccumulation and biomagnification in aquatic food web (Lavoie et al., 2013); human health concerns associated with wild food sources. |
| Σ Alk-PAHs | Raw bitumen, petroleum coke [69]; wildfire [70]. | Known mutagens and carcinogens; classified as toxic substances in Canada under Schedule 1 of the Canadian Environmental Protection Act [71]. |
| Vanadium | Petrogenic in origin; associated with stack emissions and fugitive dust from raw bitumen. | Mostly used as tracer for oil sands impacts on site; Alberta WQ guidelines only exist for irrigation and livestock water [30]. |
| Nickel | Oil sands and petroleum coke [72]. | Mostly used as tracer for oil sands impacts on site; essential metal but toxic at higher concentrations [73]. |
| Selenium | Associated with the organic (i.e., bitumen) fraction of ores [74]. | Bioaccumulation in aquatic food webs; toxic effects include deformed embryos and reproductive failure in wildlife [75]. |
| Aluminum | Crustal element associated with fugitive dust and mining activities [72]. | Known toxicity to aquatic organisms but dependent on pH, dissolved organic carbon (DOC), and hardness [76,77]. |

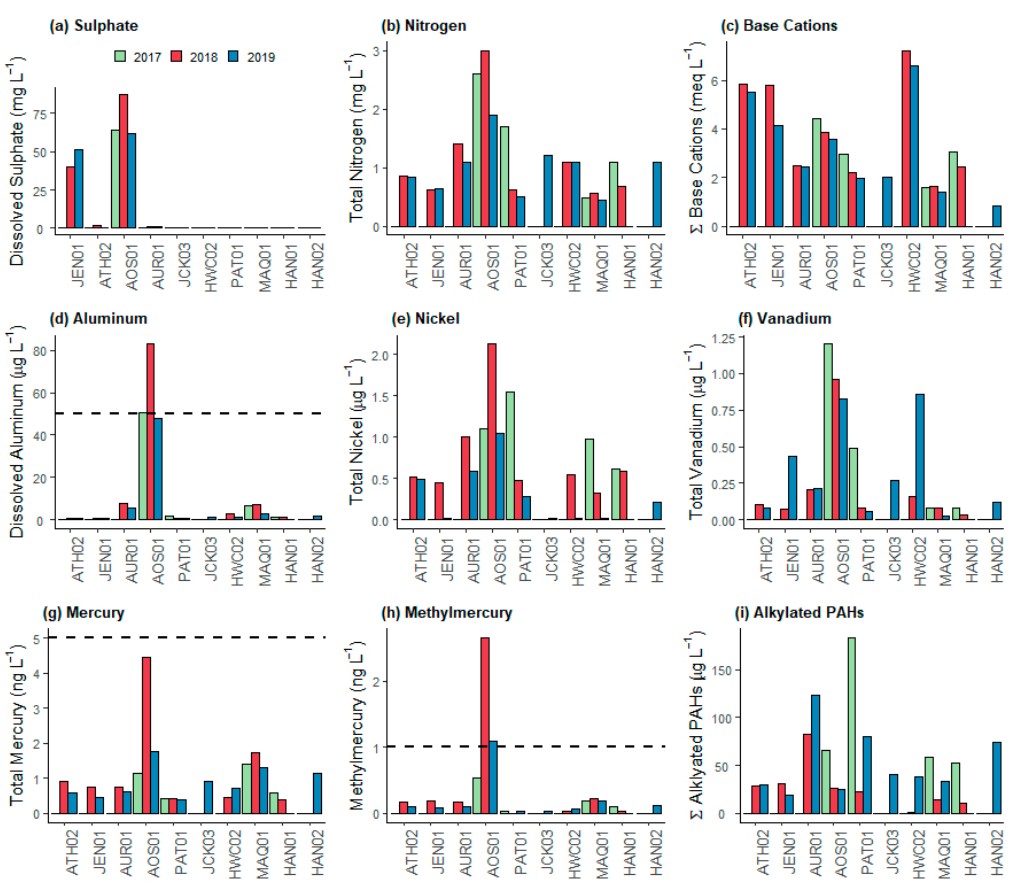

**Figure 5.** Concentrations of contaminants potentially associated with oil sands development. For sulfate (**a**), sites are ordered along the *x*-axis by increasing distance to upgrader stacks; for all other parameters, (**b–i**) sites are ordered by increasing distance to nearest oil sands surface mine. Dashed lines on panels (**d,g,h**) represent Protection of Aquatic Life chronic exposure guidelines for dissolved aluminum (50 μg L$^{-1}$), total mercury, and methylmercury (5 and 1 ng L$^{-1}$, respectively [30]).

*3.4. Benthic Invertebrates*

Three years (2017 to 2019) of aquatic invertebrate data illustrate variability across sites but not among years. Invertebrate richness and diversity (Table 4) and predator richness (Figure 6a) are detailed for shallow open water wetlands and fen complexes with open water areas. The mean invertebrate richness across sites was $35.2 \pm 5.2$ in 2017, $35 \pm 7.6$ in 2018 and $41.3 \pm 8.8$ in 2019. The mean predator richness follows a similar pattern across sites, where average richness was $10.5 \pm 2.4$ in 2017, $10.7 \pm 6.5$ in 2018, and $13.8 \pm 3.7$ in 2019. The mean invertebrate diversity among sites was consistent at $2.6 \pm 0.4$ for all three monitoring years.

**Table 4.** Aquatic invertebrate summary metrics for 2017, 2018, and 2019. Note, * denotes that 'Most Abundant Taxon' is based on proportional abundance of data summarized at LPL (lowest practical level) and does not include planktonic taxa. In some cases, there may have been a greater abundance of juvenile or damaged individuals summarized at a higher level of taxonomy (e.g., Order, Family) not represented here. Italicized text represents genus and non-italicized text represents taxonomic family.

| | **Wetland Class** | **Richness** | **Diversity** | **Most Abundant Taxon *** |
|---|---|---|---|---|
| 2017 | | | | |
| MAQ01 | Fen | 40 | 2.95 | *Caenis* (0.21) |
| AOS01 | SOW | 29 | 2.57 | *Hyalella* (0.34) |
| HAN01 | SOW | 39 | 2.13 | *Caenis* (0.50) |
| PAT01 | SOW | 33 | 2.77 | *Hyalella* (0.18) |
| 2018 | | | | |
| MAQ01 | Fen | 32 | 2.67 | Hydrozetidae (0.22) |
| HWC02 | SOW | 36 | 2.16 | *Caenis* (0.48) |
| HAN01 | SOW | 34 | 2.84 | *Chaoborus* (0.17) |
| ATH02 | SOW | 40 | 2.68 | *Caenis* (0.30) |
| AOS01 | SOW | 26 | 2.56 | *Dicrotendipes* (0.18) |
| JEN01 | SOW | 27 | 1.97 | *Caenis* (0.50) |
| AUR01 | SOW | 35 | 3.10 | Leptoceridae (0.10) |
| PAT01 | SOW | 50 | 2.79 | *Hyalella* (0.18) |
| 2019 | | | | |
| MAQ01 | Fen | 35 | 2.47 | *Stylaria lacustris* (0.33) |
| HWC02 | SOW | 49 | 2.90 | *Tanytarsus* (0.14) |
| HAN02 | Fen | 39 | 2.48 | *Chaoborus* (0.25) |
| ATH02 | SOW | 31 | 1.80 | *Hyalella* (0.56) |
| AOS01 | SOW | 48 | 2.90 | *Hyalella* (0.14) |
| JEN01 | SOW | 36 | 2.80 | *Stylaria lacustris* (0.16) |
| JCK03 | SOW | 37 | 2.32 | *Hyalella* (0.36) |
| AUR01 | SOW | 38 | 3.17 | *Psectrocladius* (0.10) |
| PAT01 | SOW | 59 | 2.99 | Baetidae (0.10) |

Neither total invertebrate richness nor predator richness varied with the human footprint (total: $r = 0.32$, $p = 0.16$; predator: $r = 0.15$, $p = 0.52$) or distance to the nearest oil sands mine (total: $r = 0.37$, $p = 0.10$; predator: $r = 0.25$, $p = 0.28$). There was a weak relationship between the human footprint in the wetland buffer and total invertebrate diversity ($r = 0.43$; $p = 0.05$) but no relationship between diversity and proximity to the oil sands operations ($r = 0.15$; $p = 0.55$). Caenis (order: Ephemeroptera) and Hyalella (order: Amphipoda) were common across sites and often the most dominant taxon by proportional abundance comprising as much as 56% of total invertebrate abundance (site ATH02 in 2019).

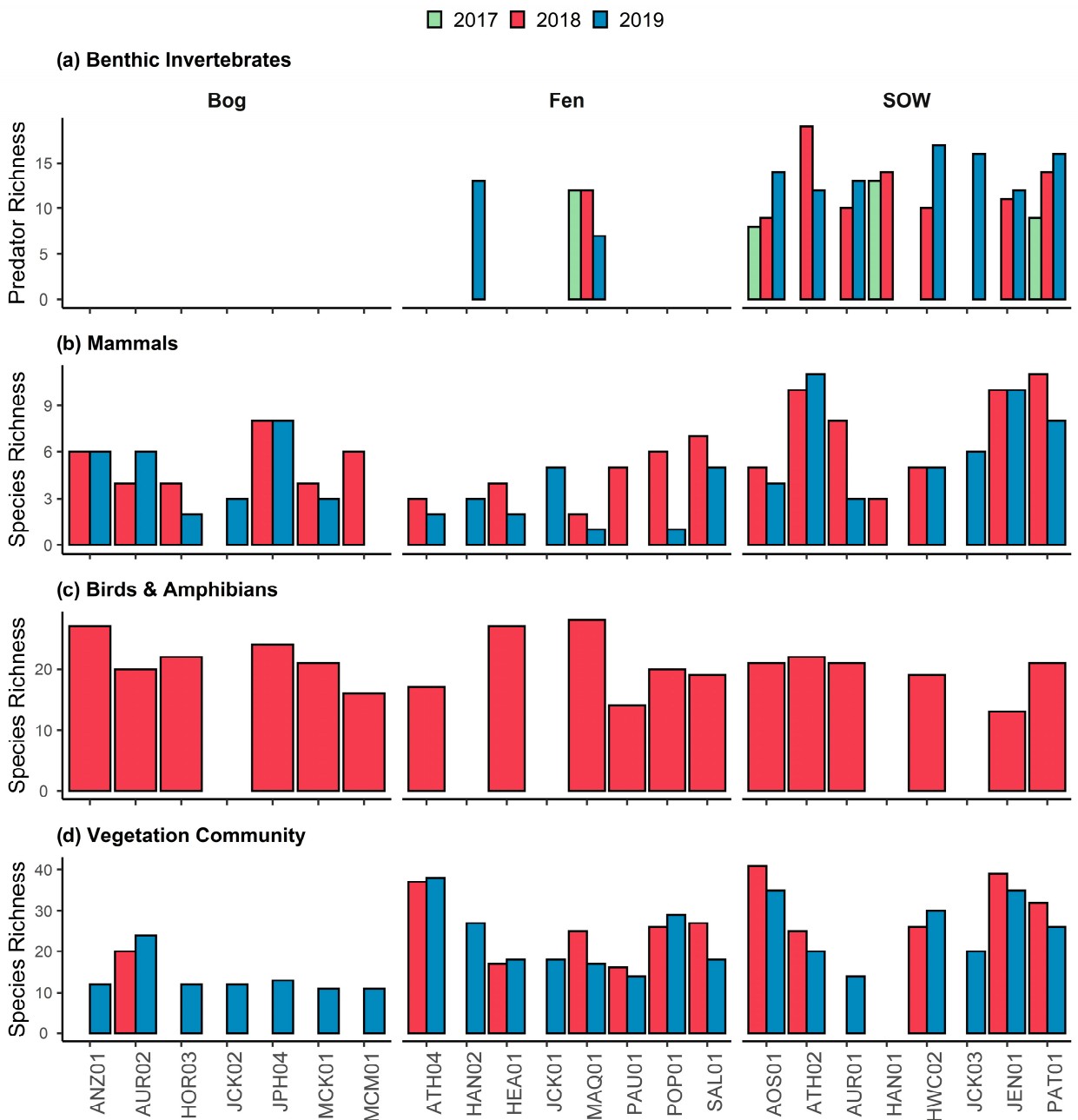

**Figure 6.** Species/predator richness for biotic indicators acquired during the pilot phase of the wetlands monitoring program: (**a**) benthic invertebrates, (**b**) acoustic data for bird observation, (**c**) wildlife camera data for mammal observation, and (**d**) vegetation communities. Data are illustrated as a function of wetland class and monitoring year.

*3.5. Wildlife*

The observed species at risk and richness (Table 5) are detailed for remote camera captures and acoustic recordings (birds and amphibians) (Figure 6b,c, respectively). The mean mammal species richness across all wetland classes in 2018 was 5.8 ± 2.6 and 4.7 ± 2.9 in 2019. In general, SOW wetlands showed the highest mammal richness of all wetland classes (7.4 ± 3.1 in 2018 and 6.7 ± 3.0 in 2019), followed by bogs (5.3 ± 1.6 in 2018 and 4.7 ± 2.3 in 2019) and fens (4.5 ± 1.9 in 2018; 2.7 ± 1.7 for 2019). Higher richness was observed across all wetland classes in 2018, with few exceptions at the site level, where 2019 richness was higher than in 2018 (Figure 6b).

**Table 5.** Remote camera and autonomous recording unit (ARU) data for 2018 and 2019 as a function of each site. Species codes are as follows: Canadian toad (CATO), common nighthawk (CONI), olive-sided flycatcher (OSFL), sharp-tailed grouse (STGR), and woodland caribou (WOCA).

| Site | Class | Remote Cameras | | | | Acoustic Recorder | |
|---|---|---|---|---|---|---|---|
| | | Richness | | Species at Risk | | Richness | Species at Risk |
| | | 2018 | 2019 | 2018 | 2019 | 2018 | 2018 |
| ANZ01 | Bog | 6 | 6 | - | - | 27 | CONI [2] |
| AUR02 | | 4 | 6 | - | - | 20 | - |
| HOR03 | | 4 | 2 | - | - | 22 | CONI [2] |
| JCK02 | | - | 3 | - | - | - | - |
| JPH04 | | 8 | 8 | - | WOCA [3] | 24 | CONI [2] |
| MCK01 | | 4 | 3 | - | - | 21 | CONI [2] |
| MCM01 | | 6 | - | WOCA [1] | - | 16 | CONI [2] |
| ATH04 | Fen | 3 | 2 | - | - | 17 | - |
| HAN02 | | - | 3 | - | STGR [2] | - | - |
| HEA01 | | 4 | 2 | - | - | 27 | CONI [2] |
| JCK01 | | - | 5 | - | STGR [2] | - | - |
| MAQ01 | | 2 | 1 | - | - | - | CONI [2] |
| PAU01 | | 5 | - | - | - | 14 | CONI [2] |
| POP01 | | 6 | 1 | - | - | 20 | CATO [4] |
| SAL01 | | 7 | 5 | STGR [2] | - | 19 | OSFL [4] |
| AOS01 | SOW | 5 | 4 | - | - | 21 | CONI [2] |
| ATH02 | | 10 | 11 | - | - | 22 | CONI [2] |
| AUR01 | | 8 | 3 | - | - | 21 | CONI [2], OSFL [4] |
| HAN01 | | 3 | - | - | - | - | - |
| HWC02 | | 5 | 5 | - | - | 19 | - |
| JCK03 | | - | 6 | - | - | - | - |
| JEN01 | | 10 | 10 | - | - | 13 | CONI [2], OSFL [4] |
| PAT01 | | 11 | 8 | STGR [2] | - | 21 | CONI [2] |

Note(s): [1] 2015 Alberta Status Designation "At risk"—East Side Athabasca Range herd; [2] 2015 Alberta Status Designation "Sensitive"; [3] 2015 Alberta Status Designation "At risk"—Richardson Range herd; [4] 2015 Alberta Status designation "May be at risk".

Mean acoustic recording species richness in 2018 was 20.7 ± 4.2 across all wetland classes. Combined birds and amphibians species richness across all sites was broadly similar, with bog having the highest richness (21.7 ± 3.7), followed by fen (20.8 ± 5.6) and SOW (19.5 ± 3.3). Concerning species at risk, common nighthawk was the most observed species at risk in 2018, detected at 13 of the 17 sites. Remote cameras observed fewer species at risk than the ARUs. Sharp-tailed grouse and woodland caribou were the only species at risk observed using cameras, whereas Canadian toad, common nighthawk, and olive-sided fly catcher were observed using ARUs.

*3.6. Vegetation*

Vegetation species richness data are presented for each site and stratified by wetland class, where data are compiled from each quadrat (*n* = 405) along each transect (Figure 6d). Bogs showed the lowest richness (13.5 ± 4.6) of all wetland classes, followed by fens (22.4 ± 8.2) and SOW (25.7 ± 8.1) (based on 2019 data). Bogs all exhibited broadly similar richness, excluding AUR02. Richness increased in some cases (e.g., POP01 and AUR01, HWC02) between 2018 and 2019; conversely, richness decreased at some sites (e.g., MAQ01, PAT01). No clear trend of increasing or decreasing richness was observed between 2018 and 2019.

## 4. Discussion

### 4.1. Addressing OSM Program Objectives

The OSM Program strives to monitor, evaluate, and report on the environmental impacts of oil sands development within the OSR of Alberta, with specific objectives related to conducting comprehensive and inclusive monitoring, which can be used to track impacts from oil sands development [78]. A challenge associated with wetland monitoring is related to identifying a suite of monitoring indicators that adequately represent wetland conditions (e.g., health and function) while simultaneously exhibiting adequate sensitivity to potential oil sands pressures such that the impacts on wetland conditions may be identified early enough to mitigate long-term degradation. The conceptual model addresses these challenges, identifying three priority wetland pressures associated with oil sands developments: land disturbance, hydrologic alteration, and contaminants.

Land disturbances include the removal of wetlands and creation of anthropogenic features and/or human alterations to wetlands. Oil sands surface mining operations typically clear nearly their entire operational lease area, whereas in situ operations are estimated to clear approximately six percent of their lease area using conservative methods [79]. Land disturbances can be subdivided into linear disturbances (e.g., seismic lines and roads) and polygonal disturbances (e.g., well pads, facilities, and forestry cut blocks) [59]. Linear features cover less area overall in the OSR but are pervasive, representing the main source of forest edges and can persist for decades [80]. Their impact on wildlife and other biota is suspected to be disproportionately larger than the area they occupy [59]. Direct land disturbances may induce secondary disturbances, such as changes to wetland vegetation, local hydrology and connectivity, and wildlife habitats, and affect human activity and presence in wetlands [40].

Sources of hydrologic alteration may result from oil sands water management activities, such as direct withdrawals (e.g., for use in oil and gas operations) and as an indirect consequence of land disturbance (e.g., infrastructure inducing changes to runoff patterns) [81]. Hydrological connectivity has been demonstrated and forecasted as increasingly vulnerable to mining activities and climate change in boreal watersheds with thinner surficial geological layers. Wetlands reliant on connectivity in these areas are predicted to become increasingly vulnerable as hydrological connectivity duration degrades [43]. Further, mines typically withdraw shallow and basal groundwater during operations to prevent groundwater seepage into mine pits, which can cause the drying of wetlands in areas adjacent to the mines. Hydrologic alterations may affect the ecological and hydrological conditions of nearby wetlands, especially those downstream of development.

Oil sands operations may emit contaminants to the surrounding environment via various sources and pathways, including upgrader stack emissions of NOx and SOx, PACs, and other contaminants from tailings ponds, coke pile fugitive dust, and vehicle fleets. In addition, the accidental release of oil sands process-affected water through groundwater pathways may affect downstream wetlands. Pollutant deposition can have harmful effects on ecosystems and biota [64,82], where impacts on the wetland state may vary based on the pollutant, receptor, and deposition pathway.

### 4.2. Wetland Indicators: Implications and Long-Term Monitoring Potential

The trends identified in the pilot program monitoring data provide the foundation for discussions related to indicator value for the assessment of oil sands developments on wetland conditions. Within the context of the three priority pressures identified using the conceptual model, primary indicators were identified for field monitoring over the long-term vision of the wetland monitoring program. Indicators may be selected for long-term monitoring if they are sensitive to oil sands development pressures, efficient, repeatable, and valued by local Indigenous communities. Some indicators may be continually observed using passive equipment installed at monitoring sites to assess long-term variability (e.g., hydrometeorological data, wildlife), whereas other indicators (e.g., water quality, benthic invertebrates, and vegetation) are monitored annually during field visits.

The implications of the pilot program results and the potential value of each indicator for long-term monitoring are discussed.

An investigation of the stressor–response relationships was justified for SOW sites as they were selected for inclusion within the site network to facilitate such analyses. Conversely, the analyses of wildlife and vegetation data are restricted to the assessment of variability within and compared to other sites between years of observation. An attempt to assess the linkages of such indicators with oil sands activities is inappropriate using pilot data because these sites were not selected with the express purpose of assessing how oil sands activities influence these ecosystem state indicators. Rather, these sites were selected opportunistically with existing projects in a colocation approach. As a result, the use of these sites for determining the linkages with oil sands development through specific pathways is limited; however, they remain suitable for assessing if a change in the ecosystem state is observable. Further, any attempt to assess how oil sands activities influence these indicators in detail will be performed in standalone investigations as part of future work.

### 4.2.1. Hydrometeorological Data

Air temperature trends are consistent between all wetland classes (Figure 3a), where any observed variability is likely related to the topographic position, latitudinal gradients, and related local climate. The differences in water temperatures likely relate to the hydrological connectivity of each wetland class and the exposure (uninsulated by peat) of open water to the ambient air temperature, solar radiation, and wind ablation effects [23,83]. Water temperatures somewhat mimic air temperature trends but vary by wetland class. Bogs show no temperature drop late in the season (unlike other classes), but non-observation is likely a limitation of the data record length. Variable surface water insulation and resulting heat latency effects, which vary between wetland classes, offer the likely cause of the observed differences in the monthly water temperature trends by class (Figure 3b). Open water environments (SOW) are uninsulated compared to peatlands (insulated with a peat layer), and, therefore, experience minimal lag in conducting heat from the air to water; peatlands experience a lag in conductance. Within peatlands, fens experience shorter lag times likely because of surface water inputs, in contrast to bogs, which are ombrotrophic.

Unique soil moisture content trends are noted for each wetland class (Figure 3c). Bogs indicate a high influx of water in May associated with spring snowmelt and groundwater thaw before the moisture content decreases in the following months as evapotranspiration dominates until precipitation becomes more substantial throughout August (Figure 4a). The observed trend demonstrates the ombrotrophic nature of bogs, highlighting precipitation and evapotranspiration as dominant hydrological drivers. In contrast, fens maintain relatively consistent soil moisture, with a decrease noted at the end of the observation period. Consistency suggests fens regulate their moisture content despite external inputs, such as large precipitation events (noted in August 2019; Figure 4b). The moisture content regulation of fens occurs through multiple hydrological pathways, including both surface and ground water connections [84]. SOW wetland soil moisture content declines throughout the observation period, whereas maximum levels are observed in May related to spring snowmelt and groundwater thaw. An August increase is likely related to precipitation events (Figure 4c). The declining trend is most severe from August to October likely due to less precipitation and changes in evapotranspiration, resulting in drawdown (observed in Figure 3d). Here, SOW moisture contents are expectedly lowest because measurements are made adjacent to the open water zone (in alignment with instrument manufacturer recommendations), which may be a transitionary wetland zone or upland. These zones are drier as they exist higher along the elevation gradient and typically exhibit a coarse, less saturated mineral substrate.

Depth to water table trends vary between SOW wetlands and peatlands. Both bogs and fens follow similar depth to water table trends; the average depths to water table indicate the water levels are at or near the ground surface (bogs) and just below (fens).

This is related to the vertical location at which sensors were installed at each site. At bogs, sensors were typically installed in hollows, which are saturated more frequently in comparison to hummocks. With improved resourcing, it is suggested that sensors should be established in hummocks and hollows to capture the surficial hydrological gradient. In general, peaks are observed in August and September, suggesting some correlation to precipitation events (Figure 4c). However, SOW wetlands indicate a stronger correlation to precipitation as observations are more variable than in peatlands.

Regardless of the wetland class, the indicator variability observed at each site (Figure 3) suggests that localized physiographic and landscape characteristics, such as natural areas (defined by the Natural Regions Committee [85]), topographic reliefs, and natural barriers (e.g., lakes and rivers) play a role in governing local climate, which is a key driver in the maintenance of wetlands [86]. This suggests that monitoring observations are nontransferable between sites, even those of the same wetland class. Examples of variability driven by these factors are most prominent at Maqua (MAQ01) and Pauciflora (PAU01), both of which experienced greater rainfall than the other sites in August 2019 (Figure 4), likely due to their higher elevation in the Stony Mountains. Similarly, interannual observations (Figure 4) indicate that the timing of precipitation events (important for wetland maintenance) may vary. This is vital contextual information for one-time or low-frequency annual observations made during field visits. Furthermore, the variability observed from two years of hydrometeorological data acknowledges that long-term records are required to identify adequate baseline conditions against which disturbances may be assessed. Long-term records are paramount for identifying conditions within the context of natural climate cycles (e.g., El Niño Southern Oscillation [87]). Ideally, hydrometeorological equipment would be established at every monitoring site; however, logistical and fiscal constraints require more effective use of equipment. Such an approach may be facilitated through the strategic deployment of hydrometeorological stations to broadly represent the climate conditions at multiple sites proximal to each other.

4.2.2. Water Quality

The analyzed water quality suite demonstrates that parameters of concern generally do not exceed the guidelines (where available) with few exceptions (i.e., aluminum and methylmercury) (Figure 5). Note: while previous studies showed elevated concentrations of selenium compared to predevelopment levels [81], total selenium was at or below the detection limit across the site network and is, therefore, excluded from the results.

The current monitoring design treats water quality parameters as covariates in the monitoring of biotic indicators, such as benthic invertebrate and vegetation communities. However, monitoring water quality as an independent indicator would require more frequent sampling during the ice-free season, including intensive monitoring during the spring freshet when a significant amount of surface runoff (containing accumulated snow-pack deposition) increases the contaminant load to receiving waters [88]. In addition, the toxicity of parameters, such as aluminum, are dependent on the in situ conditions of pH [89] and dissolved organic carbon [90]. Therefore, a static concentration of dissolved aluminum may or may not be a toxicity risk to biota depending on the variability of other parameters that require continuous monitoring (e.g., using a data sonde installed long-term).

Despite these challenges, some exceedances of water quality guidelines were detected for dissolved aluminum (in 2017, 2018) and methylmercury (in 2018, 2019) at site AOS01 (Figure 5d,h, respectively). Aluminum toxicity is largely a concern for fishes and amphibians, whereas invertebrates are more tolerant [91]. However, site AOS01 is a known location for waterfowl hunting; therefore, bioaccumulation and biomagnification of methylmercury in the aquatic food web represent potential health risks for land users. Under the EEM framework, these exceedances may act to trigger more intensive monitoring at sites of concern, including the analysis of biotic tissue to confirm their presence in the food web.

In addition to exceedances of trace metals, AOS01 also exhibits higher concentrations of total nitrogen and dissolved sulfate (Figure 5a,b) compared to the rest of the monitoring

network. Its proximity to an upgrader stack (<15 km) and a surface mine with an access road (<5 km) likely contributes to observations of increased nitrogen and sulfate. Excess nitrogen is unlikely to eutrophy open water (i.e., nonpeat-forming) wetlands as inland freshwaters in Canada are typically phosphorus-limited [92]; however, much of the region consists of wetland complexes spanning multiple classes, including bogs and poor fens that are sensitive to nutrient inputs (Wieder, 2019; Wieder, 2020). Based on water hardness, ranging from moderately hard (61 mg L$^{-1}$) to very hard (>180 mg L$^{-1}$), inputs of sulfate are low risk for toxicity to aquatic biota but still represent a potential source of acidifying deposition to the landscape. The trend of base cation concentration increasing with proximity to oil sands mines may provide some neutralizing effect to this acidifying deposition.

A major source of polycyclic aromatic compounds are piles of petroleum coke [69], which are disturbed by wind and deposit as fugitive dust. Despite ATH02 being the closest site to an open pit mine (<1 km west of Suncor Base Mine), the concentration of alkylated PAHs (Alk-PAHs) at this site was low and comparable to sites far (>50 km) outside the industrial center. Prevailing regional winds are to the east; therefore, fugitive dust input is lower than would be expected based on proximity alone. In contrast, AUR01 is directly east of the Muskeg River Mine and had the highest concentration of Alk-PAHs for the years available (2018, 2019). Some interim water quality guidelines exist for the parent compounds (e.g., acenaphthene, anthracene, fluoranthene, naphthalene, etc.), but these have not been updated since 1999 [52] and do not account for the increased persistence and toxicity associated with alkylation of the parent compounds [93]. Ecotoxicology studies are needed to understand the effects of these compounds on biota and develop Protection of Aquatic Life guidelines for the region.

Water quality is a multifaceted wetland indicator. The measured parameters did not demonstrate significant relationships with the physical variables (e.g., distance to oil sands mines, upgrader stacks, etc.) and did indicate short-term changes over the 3-year pilot phase monitoring; however, water quality remains a valuable parameter for ensuring that the water in wetlands proximal to oil sands operations remains safe (based on water quality guidelines) and to assess the impacts of oil sands contaminants. Moreover, known and/or predicted responses (Table 2) related to water quality have potentially broad implications for wetlands over large scales within the OSR.

### 4.2.3. Benthic Invertebrates

Invertebrate diversity across the monitoring network was comparable to other wetlands from the region that are considered to be relatively undisturbed (i.e., in the Peace-Athabasca Delta), with healthy invertebrate assemblages and high biodiversity [94]; however, richness was notably lower. Invertebrate communities were dominated by relatively tolerant taxa, including amphipods, Caenid mayflies, worms, and chironomids. This composition is typical of wetland habitats, which are often harsher than running waters, with large diel variability in temperature stress, pH, and dissolved oxygen, and where hypoxia is common.

Simple univariate metrics (richness, diversity) were largely unresponsive to the proxy measures of wetland disturbance (e.g., human footprint in 500 m wetland buffer, proximity to oil sands operations). This includes predator richness, which is known to respond to oil sands contamination [35]. The current wetland monitoring network includes two sites on approved mine leases, but neither contain oil sands process materials. Further work is needed to confirm which conditions of the amended wetlands were driving the effects on predators (e.g., trace metal exposure, naphthenic acid exposure, changes to pH and conductivity) to develop invertebrate-based indicators for the monitoring program.

Invertebrate diversity was weakly positively correlated with an increasing human footprint (i.e., land disturbance) in a 500 m buffer around the wetland site. The intermediate disturbance hypothesis [95] likely best explains a small increase in diversity with an increased human footprint where neither highly competitive nor highly tolerant species can dominate the ecosystem. The lack of a strong relationship between invertebrate metrics

and measures of disturbance highlights the need for indicators that incorporate several univariate metrics into an index of biotic integrity and multivariate indicators that holistically analyze the invertebrate community.

Observations of variability in communities across wetland sites and between observations made in a minimally disturbed environment, such as the Peace-Athabasca Delta, suggest that benthic invertebrate indicators are valuable for the OSM Wetlands Program.

### 4.2.4. Wildlife

The differences observed in wildlife species richness captured by camera traps (Figure 6b) may potentially be explained by the deployment time and duration of the camera traps between 2018 and 2019. However, such inferences are challenging due to a short data record (2018 and 2019) and the broad similarity observed in interannual species richness variability. Further, most camera trap captures were made during migration seasons, which is largely expected, but offers only a snapshot of presence/absence. This makes it challenging to determine with confidence if the animal is transiting through the wetland or utilizing its resources. Conversely, ARUs continuously record throughout the breeding season for amphibians and birds and provide a strong record of continual presence/absence over time. ARUs provide a stronger measure of the state of biodiversity within wetlands and their importance for breeding. The value of ARU recording is sensitive to deployment time in the boreal; if it is not deployed by late April to capture the breeding season, amphibians and birds will be missed in recordings. In this study, only a single year of ARU data was captured, which may limit its interpretation value. Theoretically, data from ARUs are quantifiable year-on-year and can be monitored long-term to establish baseline conditions, where deviations from these conditions may be indicative of a change in wetland conditions.

Discriminating species at risk by wetland class may help for the future monitoring and targeting of wetland habitat studies for species at risk, such as yellow rails. At present, geospatial data on target species acquired by ARUs have been used in combination with remote sensing technology to model wetland bird habitats in the boreal region of Alberta [96,97]; however, the available data suggest inferences of wetland conditions from long-term observations are challenging. Additional data are required to adequately assess the trends, variability, and applicability of ARUs and remote cameras within a long-term wetland monitoring program.

### 4.2.5. Vegetation

Vegetation species richness is considered a proxy for wetland conditions but varies by wetland class and within wetland class (related to the AWCS wetland form). Disturbance is expected to decrease the richness of obligate wetland species and may lead to a change in the dominant vegetation structure of transitioning AWCS wetland forms [39]. This suggests that the discrimination of obligate versus facultative species is important for assessing wetland conditions with greater confidence based on vegetation. This is also important because increased observations of invasive/non-native species (which may be introduced by disturbance) may lead to an increase in overall species richness.

Under the pilot program, vegetation observations were made along a transitionary gradient from upland towards the wetland center, resulting in samples being captured from at least one upland vegetation plot, which may elevate the observed species richness. This may have more influence on low-nutrient wetlands, which typically exhibit low species richness (e.g., bogs and poor fens). Nutrient-poor fens, such as HEA01, MAQ01, and PAU01 ($17.8 \pm 3.8$), exhibited lower species richness (for 2018 and 2019) compared to other fens monitored. The opposite is true for rich fens, such as POP01 and ATH04 ($32.5 \pm 5.9$). The similarity in the coefficients of variation of poor (0.21) and rich (0.18) fens suggests similar dispersion around the mean for species richness, indicating similar variability between the fen types. Observed absolute differences in richness suggest that the assessment of wetland conditions by vegetation may improve if the fens are subdivided by nutrient richness. No trend is noted for increasing/decreasing vegetation richness; however, the short time series

is restrictive, and long-term data records are required to provide a confident assessment. If long-term trends can be identified, they may offer insights into any wetland class and/or form transition that may occur based on species richness trajectories.

*4.3. Defining Baseline Conditions and Assessing Variability*

The confident identification of changes to wetlands and attributing if changes are a result of oil sands developments is challenging. The range of variability of wetland conditions in the OSR is currently unknown; therefore, the detection limit of disturbance at any wetland is also unknown. Multiple factors, related to oil sands disturbances, other anthropogenic disturbances (e.g., forestry), and any changes that exist within the ecosystem's natural range of variability (e.g., climate cycles, wildfires, etc.) confound identifying the source of any observed changes. Understanding baseline conditions and the natural variability that exists within these baseline conditions is required to identify the disturbance source(s) with confidence. Within a wetland ecosystem context, the "baseline" is thematic, spatial, and temporal (discussed below).

Thematic resolution relates to the resolution of wetland detail as noted within the AWCS (e.g., wetland class, form, and type). Swamps are identified as a thematic gap in the pilot program, which were not monitored because of challenges associated with their confident identification (by remotely sensed data and/or in the field) and because peatlands and open water wetlands were initially identified as a priority. To establish a meaningful baseline of wetland conditions across the OSR, swamps should be included as a wetland class for long-term monitoring. Moreover, swamps are the second most common wetland class within the OSR, estimated to occupy 16,000 km$^2$ and approximately 25% of all wetlands (by area) in the OSR [98]. The pilot program performed analysis at the wetland class level, but large variability in some indicators (e.g., vegetation) suggests that a greater thematic resolution than wetland class may be beneficial for mitigating confusion in the sources of variability. An analysis of wetlands at a high thematic resolution (e.g., wetland form or type) reduces within wetland class variability, thus narrowing variability in the baseline measurements, and provides greater confidence in identifying deviations from the baseline. For example, nutrient-rich fens exhibit greater diversity than nutrient-poor fens and bogs [99]. When analyzed together, variability is expected to be high, whereas analysis in isolation will yield unique and more constrained baseline estimates with lower variability. Monitoring at a high thematic resolution is challenging due to time and cost constraints and locating appropriate sites. However, strategically implementing different thematic resolutions where most appropriate (e.g., nutrient-rich versus -poor fens) may offer a compromise between improving our understanding of baseline conditions while balancing cost.

Spatial variation captures variability that exists in wetland conditions as a function of regional/local landscape characteristics. The 22 pilot program sites are inadequate for establishing baseline conditions because they do not appropriately capture the spatial variability of the landscape conditions (e.g., surficial and bedrock geology, topography, etc.) [45,100]. In fact, the pilot phase data are spatially limited, located within 9 km of oil sands operations. This limits the detectability of variation between sites and over time as the sites are subject to similar pressures. Change detection is further complicated because wetland sites were established after oil sands operations, meaning any changes that occurred in ecosystem states immediately after oil sands development may have normalized before wetland monitoring was initiated. An expanded monitoring network of wetland sites along a gradient of landscape characteristics that generally represent the broader OSR is required to adequately identify baseline conditions against which individual site data can be compared to identify potential indicator change and facilitate scaling techniques using remote sensing (discussed below). The identification of appropriate monitoring sites for identifying baseline conditions within the OSR requires knowledge of the regional pressures that act upon wetlands. Such knowledge facilitates the ability to control for oil sands pressures within the monitoring network by establishing sites in areas with

high sensitivity to oil sands pressures (similar to pilot site locations) versus reference areas, where little to no oil sands pressures act upon the wetlands. This will allow for the contrasting and comparison of results from high-sensitivity and reference areas to determine if anthropogenic-driven changes have occurred.

Temporal variation captures variability that exists in wetland conditions over time (e.g., seasonally, annually, etc.) and those related to changing climate and climate cycles. The three-year pilot program data are inadequate to span climate cycles and are unable to provide appropriate context for identifying change trends. The appropriate assessment of temporal variability can only be resolved through monitoring wetland conditions long-term to identify interannual variability and wetland condition responses to climate/weather cycles [101]. The observation period required to adequately identify any variability due to a changing climate and climate cycles is largely unknown. However, observation periods may be targeted based on multiple periods of known climate cycles that act within the OSR and western Canada. In the current era of observed climate change, where climate conditions are shifting upward or downward rather than fluctuating about a constant level, conventional 30-year averages may be inappropriate; shorter intervals from five to twenty years may be more appropriate [102]. Longer data records will increase confidence in answering OSM objectives related to whether changes to wetland conditions are occurring because of oil sands pressures.

The total number of wetland monitoring sites required to establish baseline conditions in the OSR is related to thematic resolution. Statistical power analyses can be used to determine the number of sites required to detect change at the statistical power level recommended [103]. As an example, statistical power analyses performed by Ficken and Rooney [104] suggest a minimum of 28 sites are required to yield a 50% detectability of vegetation richness change within wetland classes; however, the required sites may vary with each indicator. As the program develops, the science and understanding of various pressures evolve, and wetland change will become more apparent, but first, baseline conditions must be established to develop a confidence interval. With greater understanding of how conditions vary thematically, spatially, and temporally, baseline conditions can be identified with increased confidence, which, in turn, facilitate the identification of the deviation from baseline conditions with increased accuracy.

### 4.4. Scaling up Monitoring Data—Remote Sensing Strategy

Data acquired under the wetland monitoring program are inherently spatially discrete, representing a limitation with respect to the investigation and inference of implications that oil sands activities may exert on wetland ecosystems at regional scales. However, select variables were identified and monitored strategically to facilitate spatial scaling using remote sensing data. Remote sensing data have proven successful for mapping and monitoring wetland areas and abundance [38,105,106], vegetation structural and species parameters [39,107–110], and climatological variables [111,112]. In addition, remote sensing has emerged as an important tool for the effective management of natural resources through the adaptive implementation of policies and legislation [101].

In situ data can be leveraged to scale indicators using remote sensing and state-of-the-art statistical modeling techniques (e.g., deep learning) to yield spatially continuous products. Alternatively, existing remote sensing-derived data products (e.g., canopy height estimates) may be validated using in situ acquisitions. Moreover, validated products may be used as a substitute to in situ data to drive spatially continuous models. This approach, coined "lots-of-plots", was pioneered in large-scale forestry applications [113,114]. An advantage of this approach is its ability to leverage spatially discrete data with remotely sensed data as a substitute for ground truth, which, in turn, provides increased confidence in large-scale output products. Such products are valuable for assessing the effects of oil sands pressures on wetlands at scales beyond lease boundaries and with improved cost efficiencies compared to traditional in situ data acquisitions. For example, large-scale products can be utilized to assess the critical distances of sites to oil sands pressures, which

is unattainable with the current monitoring network design due to the relative sparsity of sites and their proximity to pressures. Critical distance assessment represents one of many applications that can be pursued using remote sensing [39].

## 5. Conclusions and Recommendations

This study was built upon the DPSIR model developed by Ficken, Connor [21] and identified pathways (Figure 2) and known and/or predicted responses to anthropogenic pressures from oil sands developments that act upon wetlands in the OSR of Alberta, Canada (Table 2). Moreover, this study assessed the ability to detect changes in an ecosystem state(s) using indicator data from a three-year wetland monitoring pilot program in addition to providing an overview of monitoring methods and results that provide insights for refinement as the program transitions to a long-term monitoring effort. It is important to note that this study did not determine oil sands sources of changes that may be observed in ecosystem states; this constitutes a large body of future work that is planned within the OSM Wetlands Program. The recommendations are made contextually within the pending transition of the pilot wetland monitoring program to a long-term "core" monitoring program. A primary recommendation is that the long-term program includes the monitoring of swamps, a unique wetland class that represents a large proportion of the OSR, representing a significant knowledge gap. Moreover, the long-term monitoring program will refine monitoring data indicators to those most likely to identify a change in wetland conditions, specifically hydrometeorology (for contextualizing climate conditions) and water quality, benthic invertebrates, and vegetation indicators (including those suitable for scaling using remote sensing). Wildlife indicators require further investigation to assess their value for wetland monitoring. The methods employed for long-term indicator sampling should follow rapid and reproducible protocols to ensure that sampling may be adopted by local communities, industry, and other collaborators with minimal training. This will enable independent site-level monitoring projects to be integrated and compared with the broader OSM wetland monitoring program. In transitioning to long-term monitoring, the program needs to remain adaptable to government priority changes while maintaining a data archive that remains consistent over time. It is important that any updates to monitoring methodologies ensure compatibility is maintained between contemporary and archived data.

Accommodating the need to establish baseline conditions for wetlands in the OSR, an expanded monitoring network of wetland sites is required. A proposed long-term expanded network will monitor 80 sites (20 each of bog, fen, swamp, and SOW) for reference, against which 40 high-sensitivity sites (10 of each wetland class) will be assessed for changes. Identifying reference and high-sensitivity areas in which to establish monitoring sites is under development. It is recommended high-sensitivity areas be identified at broad scales (e.g., watershed), based on the spatial concentration of the priority pressures (land disturbance, hydrological alteration, and contaminant loading; Figure 2) identified in this study. Should changes to wetland conditions be detected using this approach, further analysis may facilitate the investigation of wetlands outside of areas of high sensitivity to investigate the limit of disturbance detectability. To maximize detectability, the initial efforts in this programmatic area should be concentrated on wetlands at high risk of disturbance. Moreover, an expanded monitoring site network will provide valuable data for scaling data using remote sensing, which, in turn, may provide a holistic assessment of the OSR.

**Author Contributions:** Conceptualization, C.M., J.M., S.C. and D.C.; methodology, C.M., J.M., S.C. and D.C.; software, C.M., J.M. and S.C.; validation, C.M., J.M. and S.C.; formal analysis, C.M., J.M. and S.C.; investigation, C.M., J.M. and S.C.; resources, D.C.; data curation, C.M., J.M. and S.C.; writing—original draft preparation, C.M., J.M., S.C. and D.C.; writing—review and editing, C.M., J.M., S.C. and D.C.; visualization, C.M. and J.M.; supervision, C.M., J.M., S.C. and D.C.; project administration, D.C.; funding acquisition, D.C. All authors have read and agreed to the published version of the manuscript.

**Funding:** This research was funded by the Oil Sands Monitoring (OSM) Program under the Wetland Ecosystem Monitoring Project (WL-PD-10-2223) through Alberta Environment and Parks.

**Data Availability Statement:** The data presented are publicly available through the Oil Sands Monitoring data portal http://osmdatacatalog.alberta.ca/.

**Acknowledgments:** This work was supported by the Oil Sands Monitoring (OSM) Program under the Wetland Ecosystem Monitoring Project (WL-PD-10-2223) through Alberta Environment and Parks. This work was funded by the OSM Program and is a contribution of the program but does not necessarily reflect the position of the program.

**Conflicts of Interest:** The authors declare no conflict of interest.

## Appendix A

**Table A1.** Location and descriptions of each study site. "Description" indicates dominant vegetation species and is not an all-inclusive list. "Elev." refers to elevation above mean sea level.

| Name | Location | Elev. (m) | Description |
|---|---|---|---|
| ANZ01 (Bog) | 56.469, −111.043 | 469 | Black spruce-dominant with bog rosemary and cottongrass understory. Adjacent to train tracks in recently burned forest. |
| AOS01 (SOW) | 56.939, −111.662 | 326 | Bog birch-dominant with sedge understory. Adjacent to road and pipeline corridor. |
| ATH04 (Fen) | 56.905, −111.448 | 215 | Dwarf birch-dominant with sedge understory and scattered larch. Adjacent to winter road that transects perpendicular to flow direction. |
| ATH02 (SOW) | 56.913, −111.441 | 212 | Sedge and rush-dominant with common cattail and yellow pond lily near shoreline. Black spruce and poplar in adjacent upland. Adjacent to recent cut block (2019), within the flood plain of the Athabasca River. |
| AUR02 (Bog) | 57.272, −111.261 | 305 | Black spruce-dominant with willow and graminoid understory. Nutrient-rich wetland complex located adjacent to cut block. Complex transitions from bog to swamp. |
| AUR01 (SOW) | 57.258, −111.252 | 305 | Willow-dominant with sedge and common cattail understory. Located on reclaimed exploration area/borrow. |
| HWC02 (SOW) | 56.533, −111.322 | 421 | Willow-dominant with sedge and graminoid understory. Common cattail and rush at shoreline. Disturbed wetland/borrow pit immediately adjacent to Highway 63. |
| HAN02 (Fen) | 56.315, −111.624 | 597 | Black spruce- and larch-dominant with willow and sedge understory. Floating fen located on in situ oil sands lease, adjacent to Highway 63. |
| HEA01 (Fen) | 56.955, −111.541 | 286 | Dwarf birch-dominant with sedge understory. Scattered willow, larch, and black spruce. Wetland complex within oil sands exploration area. Road intersects connected swamp upstream. |
| HOR03 (Bog) | 56.327, −111.591 | 542 | Black spruce-dominant with bog birch, Labrador tea, and other graminoid. Within forest adjacent to pipeline corridor and Highway 63. |
| JCK02 (Bog) | 57.432, −111.254 | 265 | Black spruce-dominant with bog birch, cottongrass, and Labrador tea understory. Pristine bog, located near low-impact exploration. |
| JCK01 (Fen) | 57.442, −111.218 | 265 | Sedge- and pitcher plant-dominant in "flarks" with larch, black spruce, and bog birch in adjacent "strings". Patterned fen, located near low-impact exploration. Patterned fen is made up of strings and flarks; strings are elevated mounds, and flarks are low-lying areas between strings. |
| JCK03 (SOW) | 57.403, −111.310 | 264 | Black spruce- and willow-dominant with mixed sedge and forb understory. Low-impact SOW complex transitioning to treed swamp and shrubby/graminoid fen. |
| JEN01 (SOW) | 57.136, −111.602 | 209 | Willow, birch, sedge, and other graminoid-dominant with common cattail and horsetail near shoreline. High-impact SOW intersected by Highway 63 and pipeline corridor. Adjacent to industrial yard. |
| JPH04 (Bog) | 57.113, −111.423 | 311 | Black spruce-dominant with cottongrass, bog birch, and Labrador tea understory. Located within sandy substrate. Adjacent to active exploration and access road. |
| MAQ01 (Fen) | 56.369, −111.284 | 695 | Sedge-dominant with scattered dwarf birch. Low-impact graminoid fen hydrologically connected to lake. Large open water area at central zone. |
| MCK01 (Bog) | 57.228, −111.703 | 272 | Black spruce-dominant with bog birch, Labrador tea, and bog rosemary understory. Adjacent to Canadian Natural Resources (CNRL) Horizon Highway and exploration access trail. |
| MCM01 (Bog) | 56.627, −111.196 | 361 | Black spruce-dominant with bog birch, Labrador tea, bog rosemary, and cranberry understory. Elevated bog from surrounding fen. Surrounded by exploration access trail. |

**Table A1.** *Cont.*

| Name | Location | Elev. (m) | Description |
|---|---|---|---|
| PAT01 (SOW) | 57.511, −111.402 | 269 | Willow-dominant with scattered poplar and Jack pine. Sedge- and yellow pond lily-dominant near riparian and shoreline. Karst sinkhole within sandy upland. Adjacent to access road. |
| PAU01 (Fen) | 56.375, −111.235 | 715 | Bog birch- and sedge-dominant with scattered larch and black spruce. Located in local valley. Exhibits deep peat deposits. Intersecting road induces water pooling at northern part of fen. |
| POP01 (Fen) | 56.938, −111.549 | 294 | Larch- and black spruce-dominant with willow, bog birch, and sedge understory. Rich treed fen, adjacent to recent fire, access road, and low-impact seismic. |
| SAL01 (Fen) | 56.573, −111.276 | 372 | Saline tolerant grasses-dominant with willow and dwarf birch scattered throughout. Low-impact seismic line at north of fen. Scattered with open water areas. |

**Table A2.** Summary of specialist hydrometeorological and wildlife data recording equipment deployed at each wetland monitoring site.

| Equipment | Description |
|---|---|
| HOBO USB Micro Station Data Logger-H21-USB https://www.onsetcomp.com/sites/default/files/resources-documents/20875-E%20H21-USB%20Manual.pdf (accessed on 6 May 2023) | Processor, power, and data storage assembly unit for connected sensors. |
| HOBO Rain Gauge Data Logger-RG3 https://www.onsetcomp.com/sites/default/files/resources-documents/10241-M%20MAN-RG3%20and%20RG3-M.pdf (accessed on 6 May 2023) | Records the amount of precipitation as rainfall. |
| Soil Moisture Smart Sensor-EC5 (S-SMC-M005) https://www.onsetcomp.com/sites/default/files/resources-documents/15081-J%20S-SMx%20Manual.pdf (accessed on 6 May 2023) | Records soil moisture content and temperature. |
| HOBO Temperature/RH Smart Sensor (S-THB-M002) https://www.onsetcomp.com/sites/default/files/resources-documents/11427-O%20S-THB%20Manual.pdf (accessed on 6 May 2023) | Records temperature and relative humidity. Sensor is protected from direct radiation with a solar radiation shield (HOBO RS3). |
| Onset HOBO U20 Water Level Logger https://www.onsetcomp.com/sites/default/files/resources-documents/12315-J%20U20%20Manual.pdf (accessed on 6 May 2023) | Records pressure exerted by vertical water column (when submerged) and water temperature. Depth to water is calculated by calibrating pressure measurements against ambient barometric pressure. |
| Reconyx Hyperfire 2 Outdoor Series Camera https://www.reconyx.com/img/file/-HyperFire2UserGuide2018_04_24_v1.pdf (accessed on 6 May 2023) | Digital camera with a passive infrared motion detector and a nighttime infrared illuminator that work in combination to capture photographs of wildlife. |
| Wildlife Acoustics Song Meter SM4 https://www.wildlifeacoustics.com/uploads/user-guides/SM4-USER-GUIDE-EN20220923.pdf (accessed on 6 May 2023) | Programmable audio recorder designed for the periodic, seasonal, and long-term monitoring of wildlife bioacoustics. |

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
