# Peer review of "Oil Sands Wetland Ecosystem Monitoring Program Indicators in Alberta, Canada: Transitioning from Pilot to Long-Term Monitoring"

_water, doi:10.3390/w15101914_

Round 1
Reviewer 1 Report
The article is an interesting compendium of recommendations for the implementation of a monitoring program for the full variability of wetland oil sands ecosystems. It provides recommendations for the implementation of an appropriate monitoring program for various elements of the environment (water quality, benthic invertebrates, hydrometeorology, wildlife, vegetation). It is a comprehensive study that contains a lot of interesting data that has great potential for describing the environmental variability of these wetlands. Many reliable items of literature on the subject were used here. The introduction and formulated research goals do not raise objections, as well as the conclusions and recommendations of this work. I think that due to the importance of wetlands in the context of e.g. their role in global biogeochemical cycles and their importance for climate change (methane storage), this research can find a wide audience. They deserve to be published in the journal Water with minor adjustments, which I have summarized below.
1. Please specify the limitations of the materials and methods used, i.e. the size of measurement errors of the methods, their range of determination, the assumed error of statistical analyses, etc.
2. The materials and methods used should be specified, especially in the context of the statistical software used to create maps, as well as for laboratory determinations. Please specify the manufacturers of these devices, models, names of methods, etc. The methods of laboratory determination of water parameters can be summarized in a table and included as an appendix to the article - unfortunately, it is currently unknown what measuring range was used and how the individual parameters were determined.
3. In the water quality analysis, you refer to the limit values of specific standards. I suggest adding a list of limit values for all analyzed parameters, with reference to the relevant document. You might also consider adding these limits to the charts in the article.
4. What do the dashed lines in bar graphs (e.g. Figure 4) mean? Is it the average content of a given parameter?
5. I suggest adding tables to the discussion instead of descriptions in the text. This will make the publication more readable.
6. Please consider adding statistical analyzes to the described results, e.g. in the context of the interaction of the described parameters. The Spearman correlation matrix can be considered for this purpose. Other analyzes can also be used to characterize the variability of results, e.g. PCA, HCA, factor analysis. This could help in developing recommendations for use in later monitoring programmes.
The publication has been written in understandable, professional language. Its quality is satisfactory. When correcting, pay attention only to minor errors, such as double spaces, typos, references to individual drawings, etc.
Please correct minor typos and editorial and linguistic shortcomings, e.g.:
- Line 106: should be "focused" instead of "focussed" -
caption to Figure 1: reference to letter "d" is missing
- Line 338, 339: superscript for cubic meters should be used
Author Response
Thank you for reviewing our work. Your valuable comments have improved the manuscript. Please see attached document for detailed response.

Reviewer 2 Report
The subject of this manuscript is very interesting for readers and it needs to following minor corrections:
1- Authors must state obtained results in abstract of manuscript.
2- Authors did not show novelties of this study obviously at the end of introduction of manuscript.
Author Response
Thank you for reviewing our work. You valuable comments have improved the manuscript. Please see attached word document for detailed responses.

Reviewer 3 Report
Dear authors. That is a useful piece of information and a very good initiative to integrate sectoral and public monitoring information into the scientific process. That needs some effort to adjust the technical report to a scientific communication standard. Great work has been done by authors already; however, there is some room for improvement. Attached there are several suggestions that I hope would be useful and not very time consuming.

There are no concerns about the English language; as a non-native speaker I cannot recommend improvements. However, my concern is about using industrial jargon instead of scientific terminology. I did my best to come up with the improvement proposals.
Author Response

(The authors gave the same response as above.)
